



# Freezing from the inside. Ice nucleation in *Escherichia coli* and *Escherichia coli* ghosts by inner membrane bound ice nucleation protein InaZ.

Johannes Kassmannhuber[1,2], Sergio Mauri[3], Mascha Rauscher[1], Nadja Brait[4], Lea Schöner[1], Angela Witte[4], Tobias Weidner[5] and Werner Lubitz[1]

[1]BIRD-C GmbH, Vienna, Austria
[2]Centre of Molecular Biology, University of Vienna, Vienna, Austria
[3]Max Planck Institute for Polymer Research, Mainz, Germany
[4]Department of Microbiology, Immunobiology and Genetics, Max F. Perutz Laboratories, University of Vienna, Vienna, Austria
[5]Department of Chemistry, Aarhus University, Aarhus C, Denmark

*Correspondence to*: W. Lubitz (werner.lubitz@bird-c.at)

**Abstract**

An N-terminal truncated form of the ice nucleation protein (INP) of *Pseudomonas syringae* lacking the transport sequence for the localization of InaZ in the outer membrane was fused to N- and C- terminal inner membrane (IM) anchors and expressed in *Escherichia coli* C41. The ice nucleation (IN) activity of the corresponding living recombinant *E. coli* catalyzing heterogeneous ice formation of super-cooled water at high subzero temperatures was tested by droplet freezing assay. Median freezing temperature ($T_{50}$) of the parental living *E. coli* C41 cells without INP was detected at -20.1°C and with inner membrane anchored INPs at $T_{50}$ value between -7°C and -9°C demonstrating that IM anchored INPs facing the luminal IM site are able to induce IN from the inside of the bacterium almost similar to bacterial INPs located at the outer membrane. Bacterial Ghosts (BGs) derived from the different constructs showed first droplet freezing values between -6°C and -8°C whereas C41 BGs alone without carrying IM anchored INPs exhibit a $T_{50}$ of -18.9°C. The more efficient IN of INP-BGs compared to their living parental strains can be explained by the free access of IM anchored INP constructs to ultrapure water filling the inner space of the BGs. The cell killing rate of -NINP carrying *E. coli* at subzero temperatures is higher when compared to survival rates of the parental C41 strain.





## 1 Introduction

The conventional model for nucleation of water into embryo ice crystals by ice nucleation proteins (INPs) and further into ice describes stepwise aggregation of water molecules composed of a minimum of 275 water molecules (Pradzynski et al., 2012) by electrostatic attraction of the polar parts of water by INPs until the critical size is exceeded. (Franks, 2003; Zachariassen and Kristiansen, 2000) However, the mechanism of phase transition from

water to ice is still not fully understood and of great scientific interest.

The mechanism of ice nucleation (IN) is generally divided into homogeneous and heterogeneous nucleation. Homogeneous IN occurs without any foreign substance aiding the process of ice formation (Vali et al., 2015) and homogeneous freezing of ultra-pure water occurs only at highest super-cooled and supersaturated conditions, e.g. for atmospheric water at temperatures below -38°C. (Cziczo and Froyd, 2014) Heterogeneous freezing is defined

as IN aided by the presence of a foreign substance such as bacteria or others so that nucleation takes place at a lesser super-saturation or super-cooling than is required for homogeneous IN. (Vali et al., 2015) For homogeneous ice formation at -5°C, 45,000 water molecules are required in an embryo ice crystal though this number can drop to 600 or lower when ice nuclei are present. (McCorkle, 2009) So-called ice nuclei trigger ice formation at temperatures between 0°C and -35°C. (Lundheim, 2002; Möhler et al., 2008) Heterogeneous IN is subdivided into

four modes. (1) Immersion freezing, where the ice nuclei are incorporated in a liquid body and initiates nucleation from inside the droplet. (Cziczo and Froyd, 2014) (2) Condensation freezing, where water droplets are formed by condensation around a cloud condensation nucleus (CCN), which acts simultaneously as ice nuclei. (Fukuta and Schaller, 1982) (3) Contact freezing is caused by an ice nuclei upon contact with a droplet surface (Cziczo and Froyd, 2014; Li et al., 2009) and (4) deposition freezing where IN occurs directly from water vapor upon an ice

nuclei surface. (Vali et al., 2015; Cziczo and Froyd, 2014)

Bacterial IN proteins (INPs) bound to the outer membrane of some Gram-negative Bacteria can act as an ice nucleus *(e.g. Pseudomonas syringae, Pseudomonas putida, Erwinia herbicola, Erwinia ananas, or Xanthomonas campestris).* (Maki et al., 1974; Lindow et al., 1982a; Sun et al., 1995; Lindow et al., 1978; Phelps et al., 1986; Kawahara, 2008; Abe et al., 1989; Zhao and Orser, 1990) The well-characterized plant pathogen *P. syringae*

represents one of the most efficient bacterial ice nucleus known, initiating plant damaging ice formation at temperatures of -2°C. (Lindow et al., 1982a; Lindow et al., 1982b) INPs have the property to catalyze heterogeneous ice formation of super-cooled water by orienting water molecules into an ice-like structure. (Gurian-Sherman and Lindow, 1993; Warren and Wolber, 1991) The biological function of INPs is thought to direct ice formation into the extracellular space at warm subzero temperatures to provide adaption time for the bacterium to

freezing stress. (Lorv et al., 2014) Furthermore, the resulting osmotic imbalance due to the extracellular ice formation results in an above-average water outflow from the cell to lower the intracellular IN temperature. (Oude Vrielink et al., 2016)

As described in this concept study, freezing from the inside of the cytoplasmic space is artificial and evidently does not provide any advantage for the survival strategy of bacteria at low temperatures. Recombinant gene

technology, however, provides tools for such an artificial system. It is of specific interest to investigate IN within *Escherichia coli* and derived Bacterial Ghosts (BGs) as it has been shown recently that BGs carrying INP on their outside are able to induce IN as effective as their living counterpart. (Kassmannhuber et al., 2017) Here, in this investigation the expression of truncated forms of InaZ of *P. syringae* without its N-terminal transporter sequence for location in the outer membrane (OM) were anchored to the inner membrane (IM) or expressed in the cytoplasm

of *E. coli*. IN and freezing properties of the live bacteria and their BG derivatives were investigated and compared.



BGs are defined as empty cell envelops derived from a Gram-negative bacterium produced by controlled expression of cloned gene *E* of the bacteriophage PhiX174. E codes for a 91 aa polypeptide which forms in an oligomeric conformation a transmembrane tunnel structure through the bacterial IM and OM. (Blasi et al., 1989) Due to the osmotic pressure difference between cytoplasm and outside medium the cytoplasmic content of the bacterium is expelled. (Witte et al., 1990b; Witte et al., 1990a) The size of the transmembrane tunnel is fluctuating

between 40 - 200 nm and is located in areas of potential cell division sites predominantly in the middle of the cell or at polar sites. (Witte et al., 1992; Fu et al., 2014) The BG internal lumen is free of nucleic acids, ribosomes or other constituents whereas the inner and outer membrane structures of the cell envelope are well preserved. (Witte et al., 1993)

Bacterial INPs are composed of three characteristic protein domains, a highly repetitive central domain with

several tandem consensus octapeptides (AGYGSTLT) and non-repetitive N- and C- terminal domains. (Green and Warren, 1985; Kawahara, 2002) The relatively hydrophobic N-terminal domain (approximately 15% of the protein) is assumed to bind to phosphatidylinositol and polysaccharides and functions as membrane anchor. (Lorv et al., 2014; Kozloff et al., 1991; Kawahara, 2013; Li et al., 2012) The central domain is predicted to form ß-helical structures mimicking an ice like surface, which is able to bind water molecules and functions as template for

orientating water into a lattice structure. (Kajava and Lindow, 1993; Graether and Jia, 2001; Wolber and Warren, 1989; Pandey et al., 2016) The function of the hydrophilic C-terminal domain is not completely understood, but it is believed to have an important role in stabilizing the INP conformation. (Kajava and Lindow, 1993; Lorv et al., 2014; Kawahara, 2013) Bacterial ice nuclei are classified according to their functional nucleation temperature into three different types; type I acts at -5°C or warmer, type II from -5 to -7°C and type III below -7°C, where type III

assemblies as a core protein, type II as a glycoprotein and type I as a lipoglycoprotein. (Turner et al., 1990; Ruggles et al., 1993; Kozloff et al., 1991)

For biotechnological applications *P. syringae* the most effective bacterial ice nuclei is used by the snowmaking industry. Cell surface display system based on INPs or its N-terminal domain alone exhibit fused proteins of interest in the OM. (van Bloois et al., 2011; Fan et al., 2011; Kwak et al., 1999) Phage located reporter gene

techniques using full length INPs have been developed (Lindgren et al., 1989) for diagnostic purposes to detect *Salmonella* food contaminations. (Wolber and Green, 1990; Pattnaik, 1997)

In this study, a 162 amino acids (aa) truncated N-terminal form of InaZ (1200 aa) (Wolber et al., 1986; Green and Warren, 1985) was anchored to the inner side of the *E. coli* C41 IM, using the membrane-targeting system reported by Szostak et al. (1990). (Szostak et al., 1990) The system is based on the hydrophobic membrane spanning

domains of the truncated E (E´) and L (L´) proteins of the bacteriophages ΦX174 and MS2, respectively that localize foreign proteins to the inner membrane of the cell envelope. (Szostak et al., 1990; Szostak et al., 1997; Witte et al., 1998) The protein of interest can be anchored to the inner membrane via amino-terminal E´ sequence, the carboxy-terminal L´ sequence, or with both sequences. In this study *E. coli* cells carrying an IM anchored INP were used for ice nucleation studies and interface spectroscopy of both, the living *E. coli* cells and of corresponding

*E. coli* BGs.

To our knowledge intracellular INPs or anchored to the IM facing the cytoplasmic lumen or truncated forms of it have never been used as bacterial ice nuclei centers.



## 2. Results

### 2.1 *E. coli* and BGs carrying E´, L´ or E´-L´ anchored INP

BGs functionally carrying INPs on their outer membrane have been reported earlier (**Fig. 1A**) (Kassmannhuber et al., 2017). Here, a truncated form of InaZ was localized to the inner membrane. For the production of living *E. coli* and BGs exposing INPs to the luminal site of the inner membrane (IM) directional targeting of InaZ lacking 162 aa of the 175 aa long N-terminal domain (-NINP) via N-terminal E´ domain (E´-NINP), C-terminal L´ domain (-NINP-L´), or by fusing -NINP with both anchor peptides (E´-NINP-L´) was used (schematic presentation **Fig. 1B**). In order to express the three different forms of IM anchored -NINP fusion proteins, plasmids pBE-NINPH, pBH-NINPL and pBE-NINPHL (**Fig. 2A**) were constructed. In these plasmids expression of the different INPs is under transcriptional control of the arabinose inducible $P_{BAD}$ promoter of cloning vector pBELK. For the production of BGs *E. coli* C41cells harboring either pBE-NINPH, pBH-NINPL, or pBE-NINPHL were co-transformed with lysis plasmid pGLMivb. Growth and lysis of bacteria was monitored by measuring the optical density at 600 nm ($OD_{600}$), flow cytometry, light microscopy and via colony forming units (cfu) analysis.

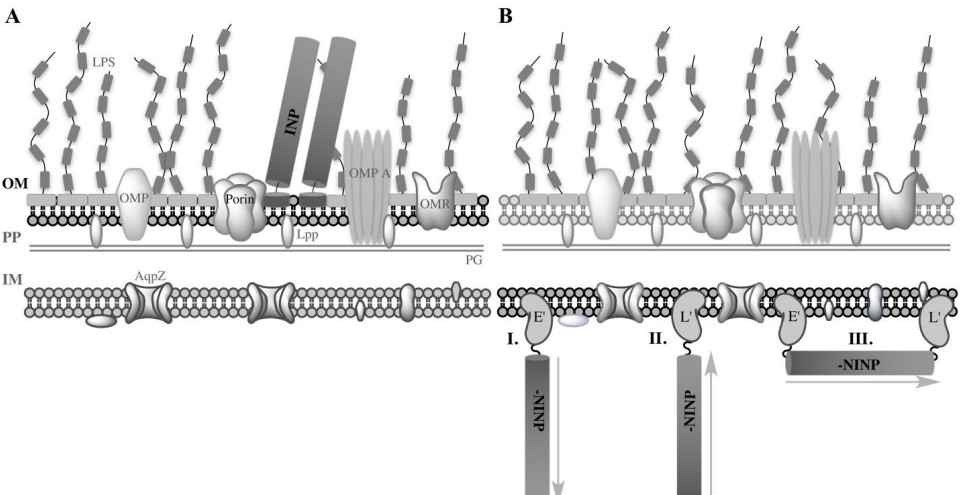

**Figure 1: BGs carrying INPs**. **A.** INPs located on BGs outer-membrane (OM) (Kassmannhuber et al., 2017). **B.** Inner membrane targeting system anchoring N-terminal truncated INP (-NINP). Illustration of -NINP anchoring modes within the BG envelope, fused to the amino-terminal sequence of the IM spanning polypeptide E´ (I.), the carboxy-terminal sequence of the IM spanning polypeptide L´ (II.), or to both sequences (III.). OM: outer-membrane; PP: periplasm; IM: inner-membrane; LPS: lipopolysaccharide; OMP: outer-membrane protein; OMR: outer-membrane receptor; PG: peptidoglycan, Lpp: Braun's lipoprotein; AqpZ: aquaporin-Z; ↑,↓: orientation of truncated InaZ fused to E´, L´ and E´-L´ anchor sequences from N- to C-terminus.

In lysis plasmid pGLMivb the expression of gene *E* is driven by the IPTG inducible $P_{TAC}$ promoter. From the beginning of growth phase expression of the different -NINP constructs in *E. coli* C41 were induced by addition of 0.2 % arabinose to the growth medium. The culture was grown at 23°C since it was reported earlier that diminished IN activity was observed when ice nucleation active (ina+) bacteria were grown above 25°C. (Chen et al., 2002; Obata et al., 1990) At an $OD_{600}$ of 0.55-0.6 (mid-log phase after about 300 min of growth) *E*-specific lysis in the recombinant bacteria was induced by addition of 0.5 mM IPTG and is referred to time point 0 min. in the growth curve (**Fig. 2B**). The efficiency of E-lysis from time of lysis induction up to 120 min lysis time was



determined by colony forming units (cfu) analysis (**Fig. 2B**). Lysis efficiency (LE) for *E. coli* C41 carrying plasmids (pBE-NINPH and pGLMivb) amounts to 99.9 %, for *E. coli* C41 carrying plasmid (pBH-NINPL and pGLMivb) a LE of 99.8 % and for *E. coli* C41 harboring plasmids (pBE-NINPHL and pGLMivb) a LE of 99.7 % was achieved. The produced BGs showed slight elongation, that is specific to the used *E. coli* C41 (**Fig. 2D**) but retained their morphological structure.

*E*-lysis of the different *E. coli* clones was also monitored online via flow cytometry (FCM) (**Fig. 2E**). The fluorescent dye RH414 staining phospholipid membranes enables discrimination of non-cellular background and DiBAC₄(3) penetrating depolarized cell membranes binding to intracellular proteins or membrane compartments signaling changes in the membrane potential were used (Langemann et al., 2010; Langemann et al., 2016). A complete switch of DiBAC-negative cells with high scatter signal (G1), to DiBAC-positive cells with a diminished

scatter signal (G2) marks the completion of protein E-mediated lysis process (**Fig. 2E**).

In order to inactivate *E*-lysis escape mutants, the cultures were further incubated at 23°C and treated with 0.17 % (v/v) β-propiolactone for another 120 min. In the final preparation of *E. coli* BGs carrying either E´-NINP (E´-NINP-BG), -NINP-L´(-NINP-L´-BG) or E´-NINP-L´ (E´-NINP-L´-BG) no viable cells were detected.

Recombinant *E. coli* C41 bacteria from time point of E-lysis induction and corresponding BGs after β-

propiolactone treatment were analyzed by Western blotting (**Fig. 2C**) using a polyclonal antiserum against -NINP (α-H-NINP). The predicted molecular mass ($M_r$) of the fusion protein E´-NINP is 109.2 kDa, $M_r$ of -NINP-L is 110.3 kDa and of E´-NINP-L´ 116.9 kDa. The full length -NINP forms were detected around their specific $M_r$. Apart from the unspecific binding of the used polyclonal antibodies with proteins derived from *E. coli*, the lower migrated bands around 75 kDa most probably represent degradation products of the above mentioned INPs.





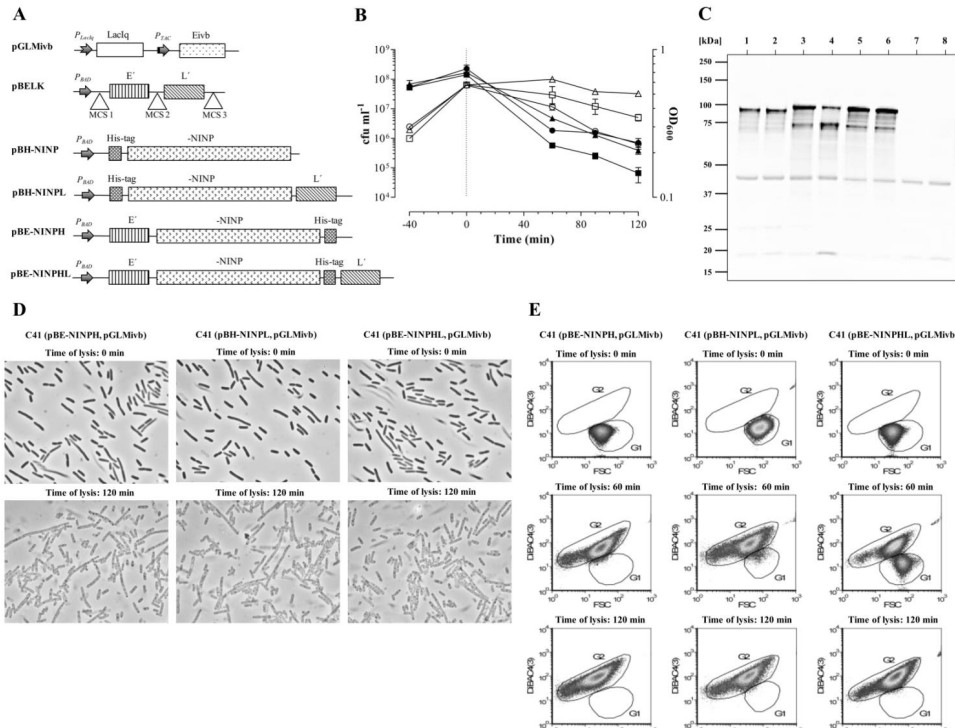

**Figure 2:** Production of BGs carrying cytoplasmic membrane located E´, L´, or E´-L´ anchored -NINP. **A.** Partial map of plasmids used for production of BGs carrying cytoplasmic membrane anchored -NINP. Eivb: C-terminal fusion of gene *E* to an in-vivo biotinylation sequence **B.** Growth and E-lysis of *E. coli* C41 (pBE-NINPH, pGLMivb), (open square) C41 (pBH-NINPL, pGLMivb) (open triangle) and C41 (pBE-NINPHL, pGLMivb) (open circle). The cultures were grown at 23°C in LBv complemented with 0.2 % arabinose to induce expression of the three distinct -NINP fusion proteins E´-NINP, -NINP-L` and, E´-NINP-L´. At time point zero (illustrated by vertical, sketched line) E-mediated lysis was induced by addition of 0.5 mM IPTG. Cfu of the corresponding cultures (closed symbols) were determined to calculate LE. After 120 min of E-lysis BG production was stopped by harvesting the cells. Data were obtained in three independent experiments. Error bars indicate standard errors. **C.** Detection of -NINP fusion proteins in *E. coli* C41 and their derived BGs. Samples were taken before induction of E-mediated lysis (Tp-0min) (lane 1, 3, 5, 7) and after β-propiolactone treatment of BG samples (lane 2, 4, 6). Western blotting was performed with rabbit anti-H-NINP serum and anti-rabbit IgG horseradish peroxidase conjugated antibodies. Lane1: C41 (pBH-NINPL, pGLMivb), expressing -NINP-L; lane 2: -NINP-L-BG; lane 3: C41 (pBE-NINPH, pGLMivb), expressing E´-NINP; lane 4: E-NINP-BG; lane 5: C41 (pBE-NINPHL, pGLMivb), expressing E´-NINP-L´; lane 6: E´-NINP-L´-BG; Lane 7: C41 control (pBAD24, pGLMivb) harvested before lysis induction (OD$_{600}$ of 0.6); lane 8: BG form of C41 (pBAD24, pGLMivb); Lines indicate molecular size marker proteins in kilodaltons (kDa). **D.** Light microscopy pictures of the recombinant *E. coli* C41 strains carrying pBE-NINPH, pBH-NINPL and pBE-NINPHL, respectively at time point of lysis induction (Tp - 0 min) and endpoint of E-mediated lysis after 120 min. **E.** Flow cytometry density dot plots during E-lysis process of *E. coli* C41 (pBE-NINPH, pGLMivb), C41 (pBH-NINPL, pGLMivb) and C41 (pBE-NINPHL, pGLMivb) strains. Online monitoring of BG production, starting at time point of lysis induction (Tp - 0min), after 60min of lysis (Tp - 60min) and end of lysis phase (Tp-120min). Dot plots illustrate fluorescence intensity with Dibac$_4$(3) versus – forward scatter (FSC); G1: living cells, G2: BGs

## 2.2 Ice nucleation activity of IM anchored –NINP

Ice-nucleating activities of *E. coli* C41constructs and BG derived version carrying either E´-NINP, -NINP-L´, or E´-NINP-L´ were determined by a droplet-freezing assay. Additionally, full *E. coli* C41 cells carrying pBH-NINP encoding a cytoplasmic N-terminal His-tagged truncated INP lacking 162 aa of the N-terminal outer-membrane



binding domain (C41-NINP) were tested for their ice-nucleating activity. For this purpose, all different -NINP
constructs in *E. coli* C41 clones harboring the plasmids mentioned above were expressed by addition of 0.2 %
arabinose to the growth medium at 23°C until the culture reached an $OD_{600}$ of 0.6.

Forty-five drops (with a fixed volume of 10 µl) from each culture suspension in ultra-pure water to be tested
(containing 5 x $10^8$ cells or BGs $ml^{-1}$) were cooled down and the number of frozen droplets at each temperature

was counted. As the first frozen droplets of *E. coli* C41 cells (median freezing temperature, $T_{50}$, of -20.1°C) and
*E. coli* C41 BGs ($T_{50}$ of -18.9°C) were detected at -14°C (Kassmannhuber et al., 2017) (**Fig. 3B**) the IN activity
of recombinant *E. coli* C41 cells described here and their BGs were monitored up to a temperature decrease of -
13°C. Out of the particular freezing temperatures of the tested droplets (**Fig. 3A**), the median freezing temperature,
$T_{50}$ (which is the temperature where 50% of all droplets are frozen) was calculated (**Fig. 3B**).

BG*s* carrying E´-NINP fusion protein (E´-NINP-BG) showed a $T_{50}$ value at -7.7°C, $T_{50}$ for BG carriers of -NINP-
L´ (-NINP-L´-BG) was determined at -8.7°C and BGs with E´-NINP-L´ anchored (E´-NINP-L´-BG) a $T_{50}$ at -9°C
was recorded. Surprisingly, non- lysed *E. coli* C41 cells carrying E´-NINP (C41 E´-NINP), -NINP-L´ (C41-NINP-
L´) and E´-NINP-L´(C41-E´-NINP-L´) (harvested before lysis induction) showed a bit higher subzero median
freezing temperatures than their E-lysed forms (**Fig. 3 A-B, Table 1**). The $T_{50}$ value of C41-NINP carrying

cytoplasmic N-terminal truncated INP was nearly identical to E´ and L´ anchored –NINP at -7.1°C. In contrast the
$T_{50}$ for *E. coli* C41 cells and *E. coli* C41 BGs were noted at -20.1°C and -18.9°C, respectively.(Kassmannhuber et
al., 2017)

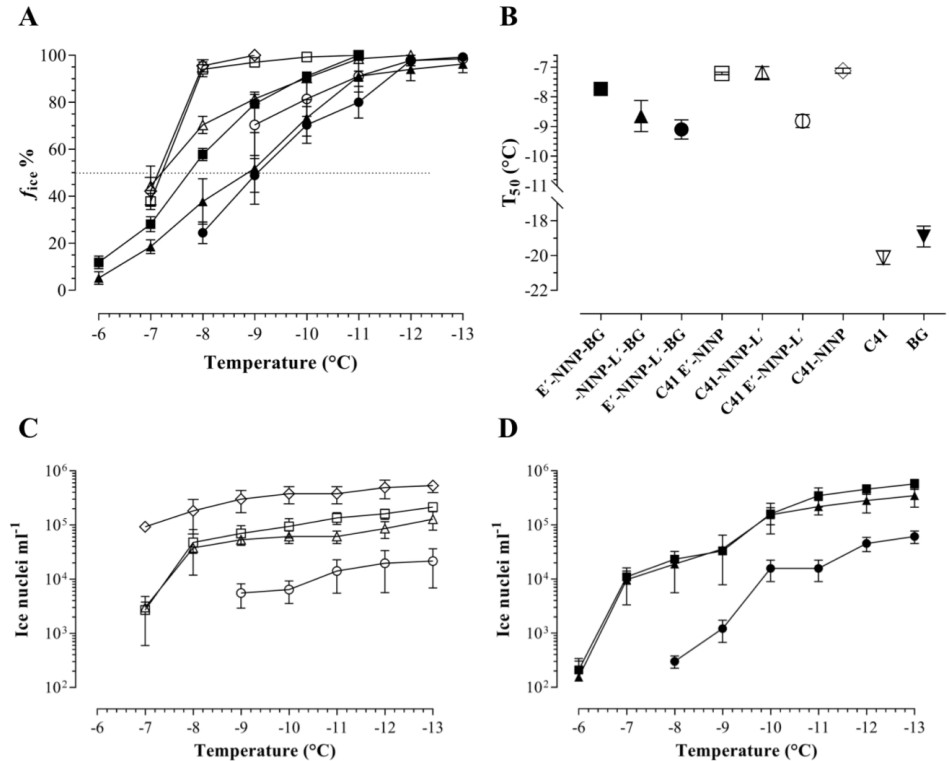

**Figure 3:** Freezing spectra (**A**), median freezing temperatures ($T_{50}$) (**B**) and cumulative spectrum of ice nucleation activity (**C, D**) of *E. coli* C41 (open symbols) and their BG derived version (full symbols): C41 E´-NINP, E´-NINP-BG (square); C41 -



NINP-L´, -NINP-L´-BG (triangle); C41 E´-NINP-L´, E´-NINP-L´-BG (circle) C41-NINP(diamond). **A.** Nucleation curve plotted as number fraction of frozen droplets in percent ($f_{ice}$ %) at any temperature. **B.** $T_{50}$, temperature where 50 % of all droplets are frozen. Forty-five 10 µl droplets of each suspension containing 5 x $10^8$ cells or BGs $ml^{-1}$ were tested by droplet freezing. $T_{50}$ of living *E. coli* C41 (C41) (▼) and its derived BG form (▽). The cumulative number of ice nuclei $ml^{-1}$ active at a given temperature of recombinant *E. coli* C41 carrying three different anchored forms of -NINP (**C**) and their BG derived

form (**D**). Average numbers are representative of three replicate assays. Error bars represent the standard errors.

As the most active ice nuclei inside a droplet determines the temperature at which the droplet freezes (Govindarajan and Lindow, 1988; Lindow et al., 1982a) a series of tenfold-dilutions from each suspension were tested to generate the cumulative IN spectra (**Fig. 3 C-D**). The nucleation frequency (NF), obtained by normalizing

the number of ice nuclei $ml^{-1}$ for the number of cells in the suspension, describes the fraction of cells in the suspension enfolding active IN sites at a given temperature (Orser et al., 1985). The average NF numbers of all tested IN active *E. coli* C41 and BGs at different temperatures are listed in **Table 1**. BGs carrying either an E´- or L´- inner-membrane anchored –NINP show a similar picture in IN spectrum with the lowest NF detected at -6°C. For E´-NINP-L´-BGs the first determinable NF was at -8°C, indicating a shift of class II INP activity ranging from

-5°C to -7°C, when –NINP is fused to E´ and L´ cytoplasmic membrane anchors. The three different *E. coli* C41 variants carrying IM anchored -NINP exhibited first detectable NF at a 1°C lower temperature compared to their recombinant BG descendant form at -7°C and -9°C, respectively. C41 E´-NINP and C41-NINP-L´ with the lowest NF detected at -7°C function as type II ice nuclei, whereas C41-E´-NINP-L´ active from -9°C can be assigned to type III active ice nuclei. C41-NINP also showed first freezing activity at -7°C, however, the NF spectrum clearly

contrasts with the others displaying significantly more nucleation sites at any temperature (**Table 1.**).

| | $T_{50}$ (°C) | **-6°C** | **-7°C** | **-8°C** | **-9°C** | **-10°C** | **-13°C** |
|---|---|---|---|---|---|---|---|
| **ina$^+$ *E. coli*** | | | | | | | |
| C41-NINP | -7.1 | n.d. | 1.9 x $10^{-4}$ | 3.6 x $10^{-4}$ | 6.0 x $10^{-4}$ | 7.6 x $10^{-4}$ | 1.1 x $10^{-3}$ |
| C41 E´-NINP | -7.2 | n.d. | 5.4 x $10^{-6}$ | 9.5 x $10^{-5}$ | 1.4 x $10^{-4}$ | 1.9 x $10^{-4}$ | 4.3 x $10^{-4}$ |
| C41 -NINP-L´ | -7.2 | n.d. | 6.0 x $10^{-6}$ | 7.6 x $10^{-5}$ | 1.1 x $10^{-4}$ | 1.2 x $10^{-4}$ | 2.6 x $10^{-4}$ |
| C41 E´-NINP-L´ | -8.8 | n.d. | n.d. | n.d. | 1.1 x $10^{-5}$ | 1.3 x $10^{-5}$ | 4.3 x $10^{-5}$ |
| | | | | | | | |
| **ina$^+$ BGs** | | | | | | | |
| E´-NINP-BG | -7.7 | 4.2 x $10^{-7}$ | 2.2 x $10^{-5}$ | 4.7 x $10^{-5}$ | 6.7 x $10^{-5}$ | 3.2 x $10^{-4}$ | 1.2 x $10^{-3}$ |
| -NINP-L´-BG | -8.7 | 3.1 x $10^{-7}$ | 2.0 x $10^{-5}$ | 3.8 x $10^{-5}$ | 7.3 x $10^{-5}$ | 3.1 x $10^{-4}$ | 7.0 x $10^{-4}$ |
| E´-NINP-L´-BG | -9.0 | n.d. | n.d. | 6.0 x $10^{-7}$ | 2.4 x $10^{-6}$ | 4.5 x $10^{-5}$ | 1.2 x $10^{-4}$ |

**Table 1:** Median freezing temperature $T_{50}$ (°C) and ice nucleation frequency (NF) at indicated temperatures. $T_{50}$ (°C): temperature where 50 % of all droplets are frozen; NF: nucleation frequency expressed as ice nuclei $cell^{-1}$; n.d.: not determinable.

**2.3 Effect of INP on water structure within bacterial ghosts**

To test the impact of INPs in the BG on the water structure, we used sum frequency generation (SFG) vibrational spectroscopy. SFG uses frequency mixing of infrared and visible laser pulses to probe the molecular structure of interfaces and is therefore ideally suited to monitor the very interface between monolayers of proteins and the surrounding water. (Perakis et al., 2016) For the signal to be measureable in the far field, inversion symmetry must

be broken, as the selection rules dictate. (Boyd, 2003) This is the case of interfaces, where inversion symmetry is necessarily broken: thus SFG is sensitive to surfaces and interfaces. Details of the theory of SFG can be found elsewhere. Here it is important to note that the selection rules also imply that increased interfacial order of




molecular groups leads to an increase of the signal strength. (Boyd, 2003)

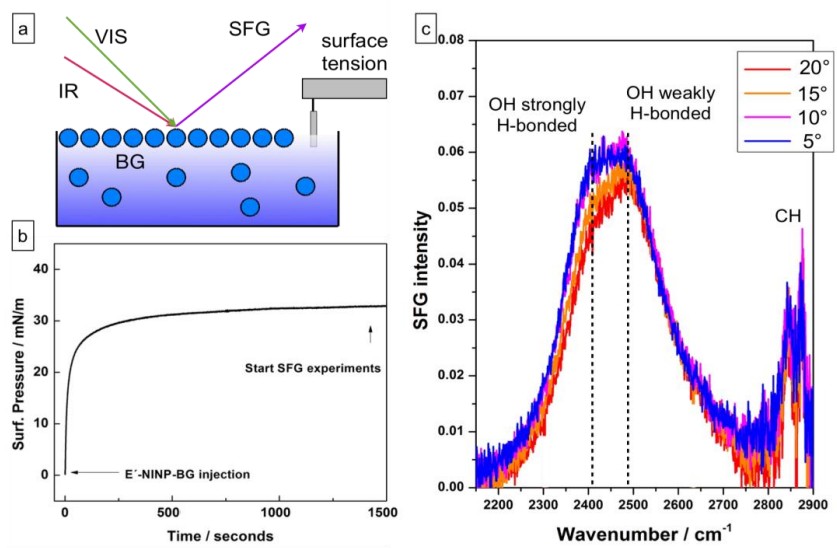

**Figure 4:** Experiments with E´-NINP-BG at the air water interface. (a) BG are assembled at the air-water interface and probed by SFG and surface tension. (b) The surface pressure relative to the water tension increases with E´-NINP-BG binding to the water surface. Injection times and time point at which the SFG experiments where started are indicated. (c) SFG spectra of the
E´-NINP-BG layer for different temperatures from room temperature to 5°C. The SFG intensity increases with decreasing temperatures.

The interaction of E´-NINP-BG with water was probed with SFG for BGs assembled at the air-water interface. For the experiments, BGs where injected into the sample trough for a final concentration of 5 x 10$^8$ ml$^{-1}$ and
allowed to bind to the water surface. Heavy water was used for the experiments to reduce the spectral width of the water region. BG adsorption at the air-water interface was verified by following the water surface tension during adsorption (Figure 4a). The data shows that equilibrium is typically reached after 1500 s. SFG spectra of the BG monolayers collected in the ssp (s-polarized SFG, s-polarized visible, p-polarized infrared) polarization combination are shown in Figure 4b. The spectra show C–H resonances between 2900 cm$^{-1}$ and 2800 cm$^{-1}$, which
are likely related to methylene units within carbohydrates, lipids, proteins side chains and other organic molecules present in the BGs. Strong modes in the O–D stretching range between 2200 and 2700 cm$^{-1}$ clearly show the presence of ordered water molecules at the bacterial membrane surfaces. (Perakis et al., 2016) Disordered water near the membrane is not detected with SFG. The room temperature spectrum (blue trace) shows a broad peak with components related to weakly and strongly hydrogen-bonded water molecules. Spectra collected at 15°C,
10°C, and 5°C showed an increase of the SFG water signal, indicating that either more water molecules or more ordered water is present at the surface. Since the water density can be assumed to be nearly constant within the general uncertainties of this experiment the latter possibility is more likely here. The signal increase is most prominent for the strongly hydrogen-bonded water species near 2400 cm$^{-1}$. Such an increase of water order near INPs for decreasing temperatures has been observed before in SFG experiments with lysed *P. syringae* within the
commercial product Snomax (Pandey et al., 2016) and for ice binding antifreeze proteins (AFPs). (Meister et al., 2014) It has also been shown that the water structure near less ice active species such as lipids, non-IN proteins, and bare air water interfaces remains unchanged in the temperature range investigated here. (Pandey et al., 2016)



5    The increase of interfacial water order close to the melting temperature is an effect that has been observed to be unique to ice-active proteins. The data clearly indicates that INP within the BGs orient water within their hydration shell when cooled to appropriate temperatures.



## 3 Discussion

In our investigation of heterogeneous immersion freezing live *E. coli* or BGs with INP bound to the IM were cooled down in ultra-pure water at subzero temperatures. Plain *E. coli* C41 or C41 BGs exhibit a $T_{50}$ of -20.1°C and -18.9°C, respectively, whereas InaZ carrying versions performed IN at a $T_{50}$ around -8°C to -9°C. In contrast to the normal INP induced freezing starting from water surrounding the bacterial particles IM -NINP induced freezing has been initiated inside *E. coli* C41 or derived BGs. The inside growing ice crystal has to find its way to the surface of the cell envelope to initiate instantaneous freezing of the water droplets monitored for freezing. It was surprising that the Gram-negative cell envelope of *E. coli* C41 did not represent a real barrier suppressing or largely delaying ice formation. The $T_{50}$ values were almost equal to IN seen in *E. coli* C41 and corresponding BGs where full length InaZ was expressed on the surface of the bacteria. (Kassmannhuber et al., 2017)

For BGs having a hole in their envelope connecting the inner aqueous lumen of the BG shell through the E-lysis tunnel with the surrounding water the enhanced IN and droplet freezing was not surprising. The SFG data also supports the exchange of internal BG water with $D_2O$ from the surrounding solvent. The ordering of interfacial water due to INPs was detected in the O–D stretching range where $H_2O$ has no SFG active modes. Since the BGs were prepared in $H_2O$, the detected $D_2O$ must have diffused into the BG. However, to explain the enhanced freezing for full bacteria there has to be postulated a water junction facilitating ice crystal growth through the triple layered envelope of Gram-negative bacteria. The OM with its open porins is like a sieve and the periplasmic space with its peptidoglycan layer is an aqueous space, but not the IM as the ultimately permeability barrier of the cell regulating the passage of substances into and out of the cell. The normal osmotic pressure difference in *E. coli* of the cytoplasm and the surrounding medium is approximately 1-3 bar (Rojas et al., 2014) and is compensated largely through the gel like structure of membrane-derived polysaccharides in the periplasmic space (PPS). Osmotic changes in *E. coli* are highly regulated using in addition to free diffusion of water through the lipid barrier of the IM also aquaporin water channels embedded within the IM connecting the PPS with the cytoplasm. Aquaporins have a role in both short- and long-term osmoregulatory responses and are required by rapidly growing bacteria. (Calamita, 2000) Hence, it is proposed that the INP expressed in the cytoplasmic lumen induce the formation of ice crystal growth from the inside of the cell to the outside shell through a connection of the cytoplasmic lumen via water channels to the outside environment.

BGs carrying E´ and L´ anchored -NINP were ice nucleation active at -6°C (1 cell of $2.4 \times 10^6$ cells and $3.2 \times 10^6$ cells, respectively) and showed a quite similar IN spectrum. This indicates that the N- or C- terminal orientation of the IM anchored -NINP is not significant and does not affect its nucleation activity. However, in contrast to live *E. coli* C41 where NF over the temperature range of -7°C to -13°C is almost similar with $T_{50}$ of -7.2°C, the same constructs in BGs differ by 0.6 and 1.5°C °C, respectively in their $T_{50}$. -NINP anchored at both terminal ends to the IM of *E. coli* C41 showed IN activity at lower temperatures when compared to E´ and L´ anchored -NINP. It seems very likely that E´-NINP-L´ anchoring could impair the mobility of the construct in the IM. Self-assembly of InaZ molecules is necessary for efficient IN bringing together the flat threonine and serine rich ice binding surface sites for mimicking the basal plane of ice. (Garnham et al., 2011) The aggregation size of INP monomers cause the different threshold temperatures (Govindarajan and Lindow, 1988) where one INP monomer is functional at -12°C, a minimum of three co-operative INP monomers are needed for ice nuclei active at -8°C and at least 50 INP monomers for activity at -2°C to -3°C. (Hew and Yang, 1992; Garnham et al., 2011) As in the E´-NINP-L´ construct IN occurs at -9°C it is strongly felt that the latter assumption seems to be probable.

The free-movement of cytoplasmic -NINP facilitates its aggregation potential leading to significant higher NF





numbers of C41-NINP compared to the membrane bound ina⁺ versions. It is surprising that even in an osmotic high concentrated cytoplasm ice nuclei/cell were more frequent in living *E. coli* C41 than in the BGs constructs with membrane anchored -NINP. The fact that components with larger volumes nucleate first (Dumont et al., 2003, 2004) and that IM -NINP BGs are connected to the water space through the E-tunnel structure it is obvious that first freezing events of such BGs can be detected earlier (-6°C for E´-NINP-BGs and -NINP-L´-BGs compared

to -7°C and -8°C for E´-NINP-L´-BG compared to -9°C) than in intact *E. coli* cells. Our findings also indicate that according to $T_{50}$ of IM anchored INPs full *E. coli* cells promote the assembling of cytosolic water molecules (water content of the cell estimated to be 70%) (Milo, 2013) into ice lattice structures somewhat better than in the BG derivatives.

In our investigation of intact bacteria carrying IM bound or free INP ice crystal formation starts in the cytoplasm

and is proceeding to the outside of the cell by crossing the cellular membrane. SFG data shows the mechanisms at the INP/water interface are similar to what has been observed for ice active *P. syringae* bacteria. The SFG signal increase observed for E´-NINP-BG is somewhat weaker compared with *P. syringae* and AFPs, which could be explained by far field cancelation effects because of the distribution of orientations of INPs within the BGs interacting with water. The presented SFG data strongly indicates that the E´-NINP-BG system indeed uses a

mechanism for internal water freezing that is very similar to natural INPs operating at the outer membrane. (Pandey et al., 2016)

Water surrounding the cells gets in contact with the cytoplasmic originated ice crystals that promote further ice crystal growth. The killing rate by ice formation in living C41-NINP and C41-E´-NINP, C41-NINP-L´and C41-E´-NINP-L´ at -20°C is accelerated injuring 100% of all recombinants at 15 minutes residence time in the deep

freezer versus approximately 80% of the controls. It is obvious that freezing from the outside and not from the inside increases the survival chances of bacteria and has therefore been favored by nature. (Lorv et al., 2014)



## 4 Material and Methods

### 4.1 Strains, plasmids and plasmid construction

All bacterial strains, plasmids and primers utilized in this study are listed in Table 2.

*E. coli* strain K-12 5-α has been used for routine cloning and *E. coli* C41 (DE3) (Lucigen) for BG production. Plasmid pBAD24 (Guzman et al., 1995), *E*-lysis plasmid pGLMivb and plasmid pBELK were obtained from BIRD-C plasmid collection. Plasmid pBELK contains an E´- L´- anchoring cassette derived from pKSEL5-2 (Szostak M., 1993) under control of $P_{BAD}$ promoter. Plasmid pEX-A2INP (Eurofins Genomics) harbors a chemically synthesized 3603 bp *ina*Z gene encoding INP of *P. syringae* S203. (Green and Warren, 1985) In lysis plasmid pGLMivb expression of the lysis gene *E* is under control of $P_{TAC}$ promoter and fused to an in vivo biotinylation sequence translational. Expression vector pBH-NINP, coding for a N-terminal His-tagged truncated INP lacking 486 bp of the 525 bp long N-terminal domain (H-NINP) has been described previously. (Kassmannhuber et al., 2017)

In order to construct a fusion between INP and the E´-anchor first a truncated INP lacking N-terminal domain (-NINP) (lacking first 485 nt) was generated. A 3171 bp fragment absent of INP N-domain sequence was produced by PCR amplification using plasmid pEX-A2INP as template and primers P1 and P2 to introduce *XbaI*- and *PstI*-restriction sites at the termini and a 6x-His tag coding sequence at 3´-end with a terminal coding sequence. The amplified PCR-product coding for -NINP-His was cloned into the corresponding sites of pBELK resulting in plasmid pBE-NINPHSLK. The 3338 bp translational fused *E´-NINP-His* PCR-fragment was amplified using pBE-NINPHSLK as template and primers P3 and P4 containing *EcoRI* and *HindIII* at terminal restriction sites. The fragment was cloned into the equivalent sites of pBAD24 to construct the E´-NINP-His anchor fusion protein expression-vector pBE-NINPH under transcriptional control of the arabinose-inducible expression system.

Using pEX-A2INP as template a 3171 bp PCR-fragment, encoding the His-NINP protein without termination codon, was obtained by PCR amplification. Primers P5 and P6 were used to introduce a 6x-His tag coding sequence at 5´-end and *EcoRI*- *PstI*- restriction sites at the terminal ends. In frame fusion of the L´- anchor and His-NINP sequence was generated by cloning the fragment into the equivalent sites of pBELK resulting in pBH-NINPLK.

By PCR using pBH-NINPLK as template and primers P7 and P8 to introduce restriction sites *EcoRI* and *XbaI* at the termini a 3366 bp PCR- fragment was obtained encoding the anchor fusion protein His-NINP-L´. The fragment was cloned into the corresponding sites of pBAD24 resulting in pBH-NINPL.

A 3165 bp PCR-fragment was generated by using P1 and P9 primers and pEX-A2INP as template to obtain the -NINP-His gene without a terminal coding sequence and 5´*XbaI* and 3´*PstI* restriction sites. The fragment was cloned into the *XbaI/PstI* sites of pBELK resulting in pBE-NINPHLK carrying the *E´- NINPH - L´* fusion gene, which translational fuses the -NINP-His sequence to the amino-terminal E´ sequence and the carboxy-terminal L´ sequence. The *E´ -NINP - L´* gene was amplified by using primers P3 and P10 to introduce *EcoRI* and *NcoI* restriction sites at the termini. The 3528 bp fragment was cloned into the corresponding sites of pBAD24 resulting in pBE-NINPHL. Figure 1 illustrates the plasmids used and constructed in this study for production of BGs carrying cytoplasmic anchored INPs facing the BG luminal site.





| Strains, plasmids and primers | Description | Source or Reference |
|---|---|---|
| *Bacterial Strain* | | |
| *E. coli* C41 (DE3) | F – ompT hsdSB (rB- mB-) gal dcm (DE3). | Lucigen |
| *E. coli* K-12 5-α | F´ *proA⁺B⁺ lacIq Δ(lacZ)*M15 *zzf::Tn10* (TetR) / *fhuA2Δ(argF-lacZ)U169 phoA glnV44 Φ80Δ(lacZ)M15 gyrA96 recA1 relA1 endA1 thi-1 hsdR17*. | NEB |
| *Plasmids* | | |
| pBAD24 | Bacterial expression vector containing the arabinose $P_{BAD}$ promoter system; restriction enzyme cloning; AmpR; ColE1 ori. | BIRD-C |
| pGLMivb | LacIq-$P_{TAC}$–*Eivb; GentR*. | (Kassmannhuber et al., 2017) |
| pEX-A2INP | $P_{LAC}$–*inaZ*, coding for InaZ, PUC Origin; AmpR. | Eurofins Genomics |
| pBELK | $P_{BAD}$-*E´-L´*-cassette for inner-membrane anchoring of proteins, KanR. | BIRD-C |
| pBH-NINP | $P_{BAD}$-*H-NINP*; AmpR. Coding for InaZ lacking N-terminal domain sequence with N-terminal His-tagged fusion. | (Kassmannhuber et al., 2017) |
| pBE-NINPHSLK | $P_{BAD}$-*E´-NINP-His*; KanR. Coding for E´ -NINP-His fusion protein. | This study |
| pBE-NINPH | $P_{BAD}$-*E´-NINP-His*; AmpR. Coding for E´ -NINP-His fusion protein. | This study |
| pBH-NINPLK | $P_{BAD}$-*His-NINP-L´*;KanR. Coding for His -NINP-L´ fusion protein. | This study |
| pBH-NINPL | $P_{BAD}$-*His-NINP-L´*;AmpR. Coding for His -NINP-L´ fusion protein. | This study |
| pBE-NINPHLK | $P_{BAD}$-*E´-NINPhis-L´*;KanR. Coding for E`-NINPHis-L´ fusion gene. | This study |
| pBE-NINPHLK | $P_{BAD}$-*E´-NINPhis-L´*;AmpR. Coding for E`-NINPHis-L´ fusion gene. | This study |
| *Primers* | *Sequence (5´→ 3´)* | *Restriction site* |
| P1: -N_INP-fwd | GTACGCTCTAGAAGTAAACACCCTGCCGGT | *XbaI* |
| P2: INP-His-rev | AAAAAACTGCAGTTATTA*ATGGTGATGGTGATGGTG*AGAGCCGGATCCCTTTACCTCTATCCAGTCATC | *PstI* |
| P3: E`-fwd | GTACCGGAATTCTTTATGGTACGCTGGACT | *EcoRI* |
| P4: His-tag-rev | GACCCAAGCTTGCAGTTATTAATGGTGATGG | *HindIII* |
| P5: His-NINP-fwd | GTACCGGAATTCACTACT*CATGCACCATCACCATCACCAT*GGATCCGGCTCTGTAAACACCCTGCCGGT | *EcoRI* |
| P6: INP-rev | AAAAAACTGCAGACTTTACCTCTATCCAGTCATC | *PstI* |
| P7: His-tag-fwd | GTACCGGAATTCCATGCACCATCACCATC | *EcoRI* |
| P8: L´-rev | GTACGCTCTAGACTTTGTGAGCAATTCGTC | *XbaI* |
| P9:INP-His1-rev | AAAAAACTGCAGA*ATGGTGATGGTGATGGTG*AGAGCCGGATCCCTTTACCTCTATCCAGTCATC | *PstI* |
| P10: L´1-rev | AACATGCCATGGCTTTGTGAGCAATTCGTC | *NcoI* |

The primer restriction sites are underlined and 6x His-tag sequence is highlighted in italic.

**Table 2.** Bacterial strains, plasmids and primers used for construction of recombinant plasmids.

## 4.2 Growth, E-lysis conditions and inactivation of bacteria

Bacterial cultures were grown in animal protein-free, vegetable variant of Luria-Bertani (LBv: 10.0 g l⁻¹ soy peptone (Carl Roth) 5.0 g l⁻¹ yeast extract (Carl Roth) and 5.0 g l⁻¹ NaCl) and supplemented with appropriate antibiotics, ampicillin (100 µg ml⁻¹), gentamycin (20 µg ml⁻¹) and kanamycin (50 µg ml⁻¹) at 37° or 23°C. To induce expression of the anchor fusion gene which is under the control of $P_{BAD}$ promoter the *E. coli* C41 cells carrying plasmid (pBE-NINPH, pBH-NINPL, or pBE-NINPHL) were grown in LBv supplemented with 0.2 % L-arabinose at 23°C. In plasmid pGLMivb the expression of lysis gene *E* is under the control of synthetic $P_{TAC}$ promoter. The bacterial lysis was induced with 0.5 mM isopropyl-D-1-thiogalactopyranoside (IPTG). Gene *E*-mediated lysis of bacteria was induced when cells reached an optical density of 0.6, and was extended for duration of 120 min. At the end of the E-lysis procedure the BGs were harvested and washed four times with 1x Vol. of sterile de-ionized water (dH₂O) by centrifugation and finally resuspended in 1x Vol. of sterile dH₂O. For





inactivation of any surviving E-lysis escape mutants from BG production representing a minor fraction of about
0.1-0.3 % the washed BGs harvest was treated with 0.17 % (v/v) of the DNA-alkylating agent β-propiolactone
(BPL, 98.5 %, Ferak) and kept for 120 min at 23°C with slow agitation. After inactivation process, the fully
inactivated cell broth consisting of BGs and inactivated surviving cells were washed twice with 1x Vol. of sterile
dH$_2$O and once with ROTISOLV® water (Carl Roth) and resuspended in 1/10x Vol. ROTISOLV® water. The
suspensions of *E. coli* C41 cells and BGs carrying cytoplasmic membrane anchored ice nucleation protein were
adjusted to a concentration of 5 x 10$^8$ cells ml$^{-1}$ determined by flow cytometry (FCM) in ROTISOLV® water.

**4.2 Determination of colony forming units (cfu)**

For cfu determination appropriate dilutions of samples in 0.85 % (w/v) NaCl solution were plated on Plate Count
Agar (PCA) (Carl Roth) using a WASP spiral plater (Don Whitley Scientific). 50 μl samples were plated on PCA
plates as triplicates. The plates were incubated at 35°C overnight and cfu data was analyzed by a ProtoCOL SR
92000 colony counter (Synoptics Ltd) the following day.

Lysis efficiency (LE) is defined as ratio of BGs to total cell counts and can be calculated with following equation
(1):

$$LE = \left(1 - \frac{cfu_{(t)}}{cfu_{(t_0)}}\right) \times 100\% \qquad\qquad (1)$$

where $t_0$ is the time point of lysis induction (LI) and t any time after LI.

**4.3 E-lysis monitoring**

BG production was monitored by measuring optical density at 600 nm (OD$_{600}$), by light-microscopy (Leica DMR
microscope, Leica Microsystems) and via fluorescence based flow cytometry (FCM) as reported earlier.
(Kassmannhuber et al., 2017) Briefly, flow cytometry was performed using a CyFlow® SL flow cytometer
(Partec). The membrane potential sensitive dye DiBAC$_4$(3) as well as the phospholipid membrane staining RH414
(both from AnaSpec) were used for fluorescent labeling. Dye RH414 was used for discriminating non-cellular
background and DiBAC for the evaluation of cell viability. Data was analyzed using FloMax V 2.52 (CyFlow SL;
Quantum Analysis) illustrating forward scatter (FSC) against DiBAC fluorescence signal (FL1, DiBAC) and
presented as 2D density dot plots.

**4.4 Western blot analyses**

Pellets of 5 x 10$^{-8}$ BGs ml$^{-1}$ were boiled in SDS gel loading buffer (1x) for 5 min and separated on Bolt™ 4-12 %
Bis-Tris Plus gel by using a XCell SureLock™ Mini-Cell electrophoresis system (Thermo Fisher Scientific). Then,
the electrophoretical separated proteins were transferred to nitrocellulose membrane (GE Healthcare) with transfer
buffer (25 mM Tris, 192 mM glycine, 20 % methanol) using a XCell II™ blot module (Thermo Fisher Scientific).
The membrane was placed in TBST (20 mM Tris-HCl, pH7.5, 100 mM NaCl, 0.05 % Tween-20) with 5 % milk
powder (Carl Roth) overnight at 4°C. Polyclonal α-H-NINP serum from rabbit was used to detect recombinant
INP. Production of antiserum has been described previously (Kassmannhuber et al., 2017). Immunodetection was
performed using α-H-NINP serum followed by α-rabbit IgG-HRP (GE Healthcare). Detection was carried out
using an Amersham ECL Western blot detection kit (GE Healthcare) and developed with ChemiDoc™ XRS (Bio-
Rad).



### 4.5 Measurement of ice nucleation activity

Ice nucleation activity was measured by droplet freezing assay using a modified device based on a method of Vali. (Vali, 1971) Out of each suspension to be tested, forty-five droplets à 10 µl of each tenfold dilutions series (ranging from $5 \times 10^8$ cells ml$^{-1}$ to $5 \times 10^4$ cells ml$^{-1}$) were tested for active ice nuclei inside the droplet at a given temperature.

The experimental setup has already been used and described in detail in a former study. (Kassmannhuber et al., 2017) Here, only a short description is given. The droplets were distributed on a sterile aluminum plate coated with a hydrophobic film, surrounded by styrofoam and covered by a Plexiglas plate for isolation. The temperature of the working plate was decreased by two two-stage Peltier elements of type TEC2-127-63-04 circuited in series. The surface temperature of the plate was measured by a small precision temperature sensor TS‑NTC‑103A (B+B

Thermo-Technik). Ice nucleation activity was tested from -2 to -13°C at a constant rate of 1°C decrements. After a 30 seconds dwell time at each temperature the Plexiglas plate was removed and the number of frozen droplets was recorded.

### 4.6 Determination of ice nucleation activity

Forty-five drops à 10 µl containing a known number of BGs or bacterial cell suspension was allowed to cool to a fixed temperature as mentioned above and the number of frozen droplets was counted. This measurement was repeated for each and every series of 10-fold dilutions to obtain statistically significant values. These different samples were compared by their median freezing temperature ($T_{50}$), which represents the temperature where 50 % of all droplets are frozen. The $T_{50}$ was calculated with the equation reported by Kishimoto et al. (Kishimoto et al.,

25 2014),

$$T_{50} = \frac{T_1 + (T_2 - T_1)(2^{-1}n - F_1)}{(F_2 - F_1)} \qquad (2)$$

where F1 and F2 are the number of frozen droplets at temperature T1 and adjacent temperature T2, and are just below and above 50 % of the total number of tested drops (n).

The cumulative number N(T) of ice nuclei ml$^{-1}$ active at a given temperature was calculated by an analogical

variant of the equation of Vali (Vali, 1971) reported by Govindarajan and Lindow. (Govindarajan and Lindow, 1988)

$$N(T) = -\ln(f) \times \frac{10^D}{V} \qquad (3)$$

where f = fraction of unfrozen droplets at temperature T, V = volume of each droplet used (10 µl), D = the number of 1:10 serial dilutions of the original suspension. N(T) was normalized for the number of cells present in each

suspension to obtain the nucleation frequency (NF) per cell by dividing ice nuclei ml$^{-1}$ through cell density (cell ml$^{-1}$).

### 4.7 Sum frequency generation spectroscopy

SFG spectroscopy was carried out according to Pandey et al. (Pandey et al., 2016) with an adapted amplifier output

of 9.5 mJ. The optical parametric amplifier was fed with 1.5 mJ. 3 µJ IR pulses were centered at 2500 cm$^{-1}$ with a full width at half maximum (FWHM) of ~800 cm$^{-1}$. Visible and IR beams were focused with 20 cm and 5 cm focal length plano-convex lenses, respectively, and overlapped at angels of 45° and 50°C on sample surface. The SFG spectra were recorded over 5 minutes.



5    **4.8 Surface Tension Experimental Details**

Surface tension has been measured using a Langmuir tensiometer (Kibron, Finland). E´-NINP-BGs at 5 x $10^8$ ml$^{-1}$ have been prepared in $D_2O$. The temperature controlled trough was thoroughly rinsed with acetone, ethanol and milli-Q water, and dried under nitrogen stream prior to measurements. The tensiometer was calibrated using pure $D_2O$ at room temperature (20 °C).




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

**Acknowledgement**

We thank Abbas Muhammad for his intensive reviewing of the manuscript.

**Competing interests**

15 The authors declare that they have no competing interests.

**Availability of data and materials**

All data generated or analyzed during this study are included in this published article.

20 **Funding**

This work was financed and supported by BIRD-C.