# Peer review of "Freezing from the inside. Ice nucleation in *Escherichia coli* and *Escherichia coli* ghosts by inner membrane bound ice nucleation protein InaZ."

_Atmospheric Chemistry and Physics, 2019_

## Referee Comment (RC1) · Anonymous Referee #1 · 22 Feb 2019

Kassmannhuber et al. present an interesting study concerning the ice nucleation activity of a protein from Pseudomonas syringae transferred to the inner membrane of Escherichia coli ghost shells. The paper is well-written and the scientific content is sound, but the structure of the paper doesn't follow the rules of ACPD and is rather confusing. The main problem, however, is that the content is not related to atmospheric chemistry and physics. Of course, one could easily find atmospheric implications in climate geo-engineering or in artificial snowing, but the authors do not even try to find a connection with the atmosphere. Therefore, I have doubts if ACP is the appropriate journal for

this research and if not a biological journal would fit much better. If the authors and the editor should decide to stay with ACP, then major changes of the manuscript are needed.

In the introduction an additional chapter is necessary, where the authors should describe the connection between biosphere and atmosphere including transport processes and what is known about bacteria in the atmosphere. They should explain how bacteria are nucleating ice, where bacterial habitats are situated and why bacteria have developed the property of ice nucleation. Also the atmospheric ice nucleation mechanisms and the protein chemistry should be explained in more detail.

The 4th chapter "materials and methods" is very fragmentary. Important information about the set-up and the evaluation of the nucleation data are missing. The authors quote Kassmannhuber et al 2017 concerning the set-up, but when reading this paper I couldn't find the crucial information concerning the set-up. All I learned was that the droplets are pretty large about 1mm3 in volume, which makes the data incomparable with other publications in the field. Also the set-up has never been compared with other set-ups using standard samples (see. e.g. Häusler et al. 2018 or Harrison et al. 2018). I also miss the cooling rates of the experiments, which has an important impact on the results. A reference concerning the SFG measurements would help to understand the spectroscopic set-up. (p.8, l.37) For the data evaluation the authors refer to Vali 1971. However, only recently Vali 2018 has been published in AMTD revisiting the former methods from 1971. The authors should update their methods accordingly and should replot their data as K(T). Eventually also ns data should be calculated for comparison.

The 2nd chapter "Results" is very much into biochemistry and are difficult to read for meteorologists, and atmospheric physicists and chemists. Figure 3 includes too less data points, which makes the T50 values very imprecise. The cooling rate would help to understand this data. The 3rd chapter "Discussion" is short and atmospheric implications are missing. A conclusion chapter has not been presented.

Minor revisions

Tables should be formatted according to the guidelines Abbreviations should be controlled for double meaning. Atmospheric scientists use e.g. INP as ice nucleating particles, INM is ice nucleating macromolecule, INA is ice nucleating activity etc.

References

Harrison, A. D., Whale, T. F., Rutledge, R., Lamb, S., Tarn, M. D., Porter, G. C. E., Adams, M. P., McQuaid, J. B., Morris, G. J., and Murray, B. J.: An instrument for quantifying heterogeneous ice nucleation in multiwell plates using infrared emissions to detect freezing, Atmos. Meas. Tech., 11, 5629-5641, https://doi.org/10.5194/amt-11-5629-2018, 2018.

Häusler, T., Witek, L., Felgitsch, L., Hitzenberger, R., and Grothe, H.: Freezing on a Chip – A New Approach to Determine Heterogeneous Ice Nucleation of Micrometer-Sized Water Droplets, Atmosphere, 9, 140, https://doi.org/10.3390/atmos9040140, 2018.

Kassmannhuber, J., Rauscher, M., Schöner, L., Witte, A., and Lubitz, W.: Functional Display of Ice Nucleation Protein InaZ on the Surface of Bacterial Ghosts, Bioengineered, 0-0, 10.1080/21655979.2017.1284712, 2017.

Vali, G.: Quantitative Evaluation of Experimental Results an the Heterogeneous Freezing Nucleation of Supercooled Liquids, J. Atmos. Sci., 28, 402–409, 1971.

Vali, G.: Revisiting the differential freezing nucleus spectra derived from drop freezing experiments; methods of calculation, applications and confidence limits, Atmos. Meas. Tech. Discuss., https://doi.org/10.5194/amt-2018-309, in review, 2018.

---

## Referee Comment (RC2) · Anonymous Referee #2 · 8 Mar 2019

The study of Kassmannhuber et al. (2019) is essentially about the ice nucleation activity of manipulated E. coli and its bacterial ghosts with the ice nucleation active protein InaZ (from P. syringae) embedded into the inner bacterial membrane. The concept of bacterial ghosts is very interesting. The study comprise a plausible procedure and a comprehensive literature research. However, I suggest to reject the paper in its current form for two main reasons:

(a) The study does not include any statements about its importance or its implication for the atmosphere. The main objective of Atmospheric Chemistry and Physics is to investigate the Earth's atmosphere including all relevant chemical and physical processes. As the link to the atmosphere is not given at all in Kassmannhuber et al. (2019), the study does not fit to the objectives of ACP.

(b) The study itself is not clearly motivated. What is the reason for the artificial generation of bacterial ghosts with embedded ice nucleation active proteins in the INNER membrane? I do not think that it is sufficient to justify the study by the fact that no one has done it before (page 3 line 41-42). An elaboration of the advantages of the new approach are essentially necessary in particular in comparison to a previous study in which the ice nucleation active protein was embedded in the outer bacterial membrane (Kassmannhuber et al. 2017).

The results of the current study are worth for publishing when the above listed points will be addressed satisfactorily. If it is not possible to relate the results from this study to atmospheric processes, I suggest to publish the work in a more microbiology related journal.

Specific comments:

Although there is no real general rule (an attempt was done in Vali et al. 2015) and things can be defined differently, the applied terminology of ice nucleation is very untypical, which makes the reading of the paper hard for experts. In the following, I list some examples:

- INP is used for ice nucleation protein instead of ice nucleating particle

- IN for ice nucleation instead of the already old term ice nuclei

- Freezing or ice nucleus spectra is usually a derived quantity (Vali 1971), it is not clear what $f_{ice}$ (Fig. 3A) means and it is defined

For good scientific practice, it is necessary to explain the uncertainties of experiments (page 8 line 10: "Error bars represent the standard errors.").

As the fraction frozen (explained in Vali et al. 2015, probably fice in the current study) and hence also T50 is a function of number of ice nucleating particles (Augustin et al. 2013), it is not appropriate to compare T50 values of different samples having different number of ice nucleating particles. It is better to use conservative quantities such as the ice nucleus spectra which is normalized to mass, number etc.

Technical corrections:

The paper is written using a very untypical structure. Please revise the paper according to instructions given at https://publications.copernicus.org/for_authors/manuscript_preparation.html . The citation style is incorrect. Usually the citation brackets have to prior to the punctuation mark. I am not sure if it is a general law, but to appreciate older studies the older paper are given first, i.e. the order is from old to new.

References:

Augustin, S., H. Wex, D. Niedermeier, B. Pummer, H. Grothe, S. Hartmann, L. Tomsche, T. Clauss, J. Voigtlaender, K. Ignatius, and F. Stratmann. 2013: Immersion freezing of birch pollen washing water, Atmos. Chem. Phys., 13: 10989-1003, doi: 10.5194/acp-13-10989-2013.

Kassmannhuber, J., M. Rauscher, L. Schoner, A. Witte, and W. Lubitz. 2017: Functional display of ice nucleation protein InaZ on the surface of bacterial ghosts, 8: 488-500, doi: 10.1080/21655979.2017.1284712.

Vali, G. 1971: Quantitative Evaluation Of Experimental Results On Heterogeneous Freezing Nucleation Of Supercooled Liquids, J. Atmos. Sci., 28: 402-09, doi: 10.1175/1520-0469(1971)028<0402:QEOERA>2.0.CO;2.

Vali, G., P. J. DeMott, O. Möhler, and T. F. Whale. 2015: Technical Note: A proposal for ice nucleation terminology, 15: 10263-70, doi: 10.5194/acp-15-10263-2015.